# How Far beyond Diabetes Can the Benefits of Glucagon-like Peptide-1 Receptor Agonists Go? A Review of the Evidence on Their Effects on Hepatocellular Carcinoma

**DOI:** 10.3390/cancers14194651

**Published:** 2022-09-24

**Authors:** Konstantinos Arvanitakis, Theocharis Koufakis, Kalliopi Kotsa, Georgios Germanidis

**Affiliations:** 1First Department of Internal Medicine, AHEPA University Hospital, Aristotle University of Thessaloniki, 54636 Thessaloniki, Greece; 2Basic and Translational Research Unit (BTRU) of Special Unit for Biomedical Research and Education (SUBRE), School of Medicine, Faculty of Health Sciences, Aristotle University of Thessaloniki, 54636 Thessaloniki, Greece; 3Division of Endocrinology and Metabolism and Diabetes Centre, First Department of Internal Medicine, AHEPA University Hospital, Medical School, Aristotle University of Thessaloniki, 54636 Thessaloniki, Greece

**Keywords:** GLP-1 receptor agonists, hepatocellular carcinoma, cancer, NAFLD

## Abstract

**Simple Summary:**

Glucagon-like peptide-1 receptor agonists (GLP-1 RAs) were drugs originally intended for the management of diabetes, while their role on the treatment of nonalcoholic steatohepatitis (NASH), and NASH-related hepatocellular carcinoma (HCC), has been at the forefront of medical investigation in recent years. This review presents a comprehensive compilation of extensive data on the putative role of GLP-1 RAs in the treatment of HCC, providing a solid foundation for further clarification of the molecular pathways involved.

**Abstract:**

Hepatocellular carcinoma (HCC) is characterized by poor survival rate and quality of life, while available treatments remain generally limited. Glucagon-like peptide-1 receptor agonists (GLP-1 RAs) originally emerged as drugs for the management of diabetes, but have also been shown to alleviate cardiorenal risk. Furthermore, they have demonstrated a wide range of extraglycemic effects that led to their evaluation as potential therapies for a variety of diseases beyond diabetes, such as obesity, neurogenerative disorders and nonalcoholic fatty liver disease. Given the presence of the GLP-1 receptor in hepatocytes, animal data suggest that GLP-1 RAs could regulate molecular pathways that are deeply involved in the genesis and progression of HCC, including inflammatory responses, tumor cell proliferation and oxidative stress, through direct and indirect effects on liver cells. However, future studies must assess several aspects of the benefit-to-risk ratio of the use of GLP-1 RAs in patients with HCC, including co-administration with approved systemic therapies, the incidence of gastrointestinal side effects in a high-risk population, and weight loss management in individuals with poor nutritional status and high rates of cancer cachexia. In this narrative review, we discuss the potential role of GLP-1 analogs in the treatment of HCC, focusing on the molecular mechanisms that could justify a possible benefit, but also referring to the potential clinical implications and areas for future research.

## 1. Introduction

Glucagon-like peptide-1 (GLP-1), a multifaceted 30-amino acid hormone produced in the L-cells of the small intestine and proximal colon, predominantly acts as an incretin in the β pancreatic cells, potentiating glucose-dependent insulin secretion [1]. Among the numerous metabolic effects of GLP-1 are increased satiety, decrease of gastric emptying and increased bowel motility [2]. The enzyme dipeptidyl peptidase-4 metabolizes approximately 75% of GLP-1 leaving the gut, while the liver further degrades GLP-1, leading to less than 5% of GLP-1 eventually reaching the systemic circulation, raising the question of whether GLP-1 actions are conveyed via sensory neurons in the gastrointestinal tract and the liver expressing the GLP-1 receptor [3]. GLP-1 binds to glucagon-like peptide-1 receptor 1 (GLP-1R), a G-protein-coupled receptor (GPCR), which is expressed in many tissues including the central nervous system [4], α, β, and d cells of the pancreatic islets [5], the heart [6], and the gastrointestinal tract [7].

The incretin effect, mediated by the actions of GLP-1 and another intestinal hormone, the gastric inhibitory polypeptide, is known to play a key role in the preservation of normal glucose homeostasis in humans. More specifically, orally digested glucose provokes a significantly stronger insulin response compared to that resulting from intravenous administration of the same amount of glucose load [8]. Several works have demonstrated that in people with type 2 diabetes (T2D), the mechanisms that control the incretin effect are defective, leading to impaired regulation of insulin and glucagon secretion and exaggerated plasma glucose excursions after meals [9,10]. Furthermore, functional deficits in GLP-1 signaling have been documented in people with obesity, related to dysregulation of homeostatic and hedonic control of energy balance, which undermines the ability to lose weight and maintain weight loss, thus being a significant driver of the obesity phenotype [11].

The importance of the incretin effect in the pathogenesis of T2D and obesity is highlighted by the fact that exogenous administration of drugs that mimic endogenous GLP-1 actions, namely GLP-1 receptor agonists (GLP-1 RAs), has the ability to lower blood glucose levels and promote weight reduction by correcting multiple pathophysiological abnormalities of the two entities [12]. Recent guidelines for the management of T2D advocate the use of GLP-1 RAs in various stages of the disease course, such as in patients who are not adequately controlled with metformin, those with established atherosclerotic cardiovascular disease (CV) disease or at high CV risk regardless of the quality of glycemic control, those in whom weight loss is desirable and as the first injectable treatment in patients who do not meet glycemic goals with oral therapies [13]. The potential of GLP-1 RAs to effectively reduce blood glucose with minimal risk of hypoglycemia and simultaneously promote weight loss has been a breakthrough in the modern diabetes management; however, their revolutionary impact on clinical practice is based on data showing that they can alleviate cardiorenal risk by exerting anti-inflammatory and anti-atherosclerotic effects on the heart, vessels, and kidneys, improve the lipid profile and mitigate oxidative damage and endothelial dysfunction [14,15,16]. Furthermore, they have been shown to possess neuroprotective, anti-infectious and metabolic regulatory properties [17].

Although originally intended for the treatment of T2D, the wealth of mechanism of action of these drugs prompted their evaluation as potential therapies for a number of diseases beyond diabetes. Increasing evidence indicates that GLP-1 analogs can cross the blood–brain barrier and have direct effects on brain cells, with preliminary data from animal models demonstrating potential benefits in neurodegenerative disorders such as Alzheimer’s and Parkinson’s disease [18,19]. Data from cardiovascular outcome trials (CVOTs) conducted with various members of the class in people with T2D show that the reduction in the risk of major adverse cardiac events observed with GLP-1 RA therapy is primarily driven by a consistent decrease in stroke rates [20]. Liraglutide and semaglutide have been licensed for the management of obesity regardless of diabetes status [21], while several studies have produced promising results on the effects of the category in women with polycystic ovary syndrome [22]. Furthermore, the role of GLP-1 RAs in the treatment of nonalcoholic fatty liver disease (NAFLD) [23] and nonalcoholic steatohepatitis (NASH) [24], the aggressive form of NAFLD, has been the focus of medical research in recent years, as the high prevalence of NAFLD among people with T2D and obesity, and the pathophysiological links between these entities, raise the expectation that patients can derive common benefits from GLP-1 RA therapy [25]. Preliminary evidence suggests positive effects of another class of antidiabetic agents, sodium-glucose cotransporter 2 inhibitors, on the mechanisms leading to the development of hepatocellular carcinoma (HCC) [26]. However, the effects of GLP-1 analogs on carcinogenesis remain poorly understood, although from a strictly mechanistic perspective, these agents have the potential to interfere with key mechanisms involved in tumorigenesis and progression.

In this narrative review article, we discuss the putative role of GLP-1 RAs in the treatment of patients with hepatocellular carcinoma, and especially NASH-related HCC, focusing on the molecular pathways that could justify a possible benefit.

### 1.1. The Role and Effect of the Glucagon-like Peptide-1 Receptor Agonists on Liver Hepatocytes and Immune Cells

A few studies have reported the presence of GLP-1Rs on hepatocytes, with Wheeler et al. initially demonstrating that the GLP-1R gene was expressed at lower levels in rat liver [27]. Campos et al. reported that mRNA transcripts for the GLP-1Rs were detected in mice liver, supporting the hypothesis that GLP-1 may also modulate glucose disposal, possibly at the level of the hepatocyte [28]. Moreover, Gupta et al. demonstrated that both GLP-1R mRNA and protein were present on the plasma membrane of primary human hepatocytes and HuH-7 cells, and that GLP-1Rs were internalized in the presence of agonist stimulation by GLP-1 or exendin-4 (Ex-4) [29]. They also provided evidence that Ex-4 independently increased the phosphorylation of 3-phosphoinositide-dependent kinase-1 (PDK-1), protein kinase B (AKT) and protein kinase C (PKC-ζ) in HepG2 and HuH-7 cells, reducing their triglyceride stores and, consequently, hepatocyte steatosis, providing a plausible mechanism by which Ex-4 bypasses AKT activation in patients with hepatic insulin resistance. Along the same line, it has been established that GLP-1 and Ex-4 stimulate hepatocyte cyclic adenosine monophosphate (cAMP) production, as well as inhibiting the mRNA expression of stearoyl-CoA desaturase 1 and genes linked with fatty acid synthesis in primary rat hepatocytes, haulting hepatic steatosis in ob/ob mice by the improvement insulin sensitivity [30].

In a recent in vitro model of steatotic HepG2 cells, results demonstrated that Ex-4-induced the activation of GLP-1R, and reduced oleic acid-mediated steatosis in HepG2 cells by attenuating fatty acid uptake and transport via fatty acid-binding protein-1 gene downregulation, while the effect of Ex-4 was found to be Wnt/β-catenin pathway-dependent [31]. In the same in vitro model, the correlation of long non-coding RNAs (LncRNAs) with steatosis and the beneficial effect of Ex-4 were also reported, as functional enrichment analysis demonstrated that Ex-4 mediated many important pathways, such as fatty acid and pyruvate metabolism, as well as insulin, PPAR, Wnt, TGF-β, mTOR, VEGF, NOD-like, and Toll-like receptor signaling pathways [32]. In addition, at least in the pre-clinical stage, a dual glucagon-like peptide-1 receptor/glucagon receptor (GLP-1R/GCGR) agonist inhibited liver fibrosis via blocking the activation of the pro-inflammatory nuclear factor kappa B/NF-kappa-B inhibitor alpha (NFκB/IKBα) signaling pathway, as well as the c-Jun N-terminal kinase (JNK)-dependent initiation of hepatocyte death. Moreover, the aforementioned agent also attenuated hepatic stellate cell activation via the inhibition of TGF-β, as well as the downstream Smad signaling pathways, especially in CCl4-and *S. japonicum*-related liver fibrosis [33]. Furthermore, liraglutide suppressed hepatic steatosis via the restoration of autophagy, in particular the GLP-1R- transcription factor EB (TFEB)-mediated autophagy-lysosomal pathway [34]. Exendin-4 was further demonstrated to potentiate hepatic ATP-binding cassette transporter A1 (ABCA1) expression, which is known to influence hepatic cholesterol transportation and, in turn, partake in fatty liver disease, and to attenuate lipid accumulation via the Ca^2+^/calmodulin (CaM)-dependent protein kinase kinase/CaM-dependent protein kinase IV (CaMKK/CaMKIV) pathway [35]. Finally, in a diet-induced mouse model of NASH, exenatide had a beneficial effect on the lipid metabolism and attenuated the hepatic lipid byproducts that were correlated with insulin resistance and lipotoxicity, while at the same time decreased the expression of hepatic lipogenic genes (Srebp1C, Cd36) and genes associated with liver inflammation and fibrosis (Tnfa, Timp1) [36].

Another study by Svegliati-Baroni et al. provided further evidence regarding the expression of GLP-1Rs in human hepatocytes and in HepG2 cells, demonstrating that GLP-1R expression was reduced in patients with NASH, while GLP-1R activation reduced the levels of fatty acids in hepatocytes from rats with NASH by 30%, and GLP1-R activation by exenatide inhibited c-Jun N-terminal kinase (JNK) phosphorylation in NASH-hepatocytes, with the effect being peroxisome proliferator-activated receptor (PPARγ)-dependent [37]. The effect of exenatide on hepatic intracellular signaling was further evaluated, reaching a conclusion that GLP-1Rs could partake in hepatic insulin resistance, and that long-acting GLP-1R activators such as exenatide could be suitable candidates for the treatment of NASH and, consequently, NASH-related HCC. Most importantly, Yokomori et al. demonstrated both in normal and NASH human liver biopsies that GLP-1Rs were located at basolateral hepatocytes and areas with macrovesicular steatosis in the lipid microdomains, and were expressed in monocytes/macrophages infiltrating hepatic sinusoids [38]. Regarding the potential anti-inflammatory effect of GLP-1 RAs via the NF-κB pathway, exenatide was demonstrated to attenuate human arterial vascular smooth muscle cell calcification via the inhibition of the NF-κB/receptor activator of nuclear factor-κB ligand signaling, by reducing the p-NF-κB levels [39].

### 1.2. Glucagon-like Peptide-1 Receptor Agonists’ Effects on Nonalcoholic Fatty Liver Disease and Steatohepatitis

NASH, the progressive form of NAFLD that can lead to cirrhosis and HCC formation, is rapidly becoming the leading cause for end-stage liver disease or liver transplantation [40]. In Japan, liver-associated diseases, such as cirrhosis and HCC, have become the third leading cause of death (9.3%) in patients with T2D [41]. According to a meta-analysis of eighty clinical trials, the global prevalence of NAFLD, NASH, and advanced fibrosis in patients with T2D were 55.5%, 37.3% and 17.0%, respectively [42]. Moreover, the beneficial role of GLP-1 RAs on patients with T2D and NASH was demonstrated in a meta-analysis by Zhu et al., concluding that GLP-1 RAs greatly reduced intrahepatic adipose tissue, weight mean difference, subcutaneous adipose tissue and visceral adipose tissue, while the levels of alanine aminotransferase (ALT), aspartate aminotransferase (AST), body weight, body mass index, waist circumference, fasting blood glucose, hemoglobin A1c (HbA1c), total cholesterol and triglycerides were significantly decreased [43]. The efficacy and safety of liraglutide, a long-acting GLP-1 analogue, was evaluated in patients with NASH, providing evidence that liraglutide was safe, well tolerated and mediated the histological resolution of NASH [44]. The aforementioned findings were in line with a similar study of Japanese patients with NASH, which demonstrated that liraglutide can greatly improve liver function and histological hallmarks of NASH in glucose-intolerant patients [23]. Furthermore, liraglutide has been demonstrated to decrease hepatic inflammation and injury in advanced lean NASH [45], while Yu et al. provided evidence that liraglutide ameliorates NASH by inhibiting NOD-, LRR- and pyrin domain-containing protein (NLRP3) inflammasome and pyroptosis activation via mitophagy [46].

Another meta-analysis of eleven randomized-controlled trials (RCTs) evaluating the utilization of GLP-1 RAs for the management of NAFLD and NASH reached a conclusion that, in comparison with placebo or reference therapy, the use of GLP-1 RAs for a median of 26 weeks was correlated with notable reductions in the absolute percentage of liver fat content and serum liver enzyme levels, as well as with considerable histological resolution of NASH without the worsening of liver fibrosis [47]. Similarly, a meta-analysis of RCTs and observational studies by Dong et al., demonstrated that GLP-1 RA therapy greatly reduced liver histology scores for steatosis, lobular inflammation, hepatocellular ballooning and fibrosis, which are considered hallmarks of NASH [48]. Along the same line, a meta-analysis of twelve RCTs regarding the effect of GLP-1 RAs on liver enzymes and the lipid profile of patients with NAFLD showed that the treatment with GLP-1 RAs reduced ALT, gamma-glutamyl transferase (γGT) and alkaline phosphatase concentrations, suggesting that GLP-1 agonists may have beneficial impact on the treatment of NAFLD, or at least prevent its progression [49]. Finally, a meta-analysis was conducted by Dai et al. evaluating the efficacy of GLP-1 RAs in patients with metabolic-associated fatty liver disease (MAFLD), reaching a conclusion that, regardless of T2D, GLP-1 RAs improve liver injury and metabolic disorder in patients with MAFLD [50]. The potential hepatoprotective effects of GLP-1 RAs for the treatment of NAFLD or NASH include a significant improvement in hepatic fat content as assessed by imaging techniques (for all GLP-1 RAs), alongside histological resolution of NASH without worsening or improvement of histological fibrosis (data available only for liraglutide and semaglutide) [51] (Table 1).

### 1.3. The Progression of Nonalcoholic Steatohepatitis to Hepatocellular Carcinoma

The prevalence of NAFLD in the general population was reported to be about 25% in the United States, while the estimated prevalence of NASH was 5–6% in adults, and the incidence of NASH-associated HCC was reported to be 0.44 per 1000 person-years [52]. It is well established that several different etiological factors contribute to the development of HCC. NAFLD is characterized by hepatic steatosis with chronic hepatic accumulation of lipids leading to lipotoxicity—a key mechanism in NASH pathogenesis. Hepatic lipid deposition leads to metabolic reprogramming, which is characterized by a combination of cellular metabolic alterations and accumulation of toxic lipid metabolites that favor endoplasmic reticulum stress, oxidant stress and hepatocellular inflammasome activation, thus leading to the development of liver tumorigenesis. In addition, NAFLD involves several immune cell-mediated inflammatory processes, especially when reaching the stage of NASH, during which inflammation becomes an integral part of disease progression [53].

The hepatic immune cell landscape is diverse, further evolving during NASH with direct implications for disease severity. Current evidence regarding the role of immune cells in the onset and progression of NASH are not robust enough and, in many cases, details on the molecular mechanisms involved are lacking. However, a wide variety of immune cells are observed in human biopsies and mouse models taking part in the inflammatory environment which promotes hepatocyte injury and liver fibrosis, further potentiating liver disease [54]. These immune cells include neutrophils, monocytes and innate-like T cells, such as invariant natural killer T (iNKT) cells, mucosal-associated invariant T cells and γδ T cells, and conventional CD8+ T cells and CD4+ T cell subsets. IgA+ plasma cells can potentiate HCC progression through their immunosuppressive effects by enhancing CD8+ T cell exhaustion. Moreover, fatty acid-mediated cytotoxicity causes the loss of CD4+ T cells, inhibiting their tumoricidal activity and, in turn, favoring the growth of NASH-related HCC. Finally, CD8+ T cells and, most importantly, the hepatic steatosis-induced CXCR6+ subset, were associated with liver injury and potentiated the transition of NASH to HCC via the secretion of pro-inflammatory cytokines and direct hepatocyte killing in a tumor necrosis factor (TNF)-dependent manner [55].

Expanding our knowledge on the mechanisms by which immune cells contribute to the pathogenesis of NASH would aid the design of innovative drugs, as current therapeutic options for NASH are limited. The exact mechanism by which this mixture of inflammatory microenvironment, aberrant metabolism and ongoing liver regeneration contributes to DNA instability and HCC is still poorly understood. Recently, it was reported that the prognostic liver signature (PLS)-NAFLD predicted incident HCC over up to 15 years of longitudinal observation. High-risk PLS-NAFLD was correlated with IDO1+ dendritic cells and dysfunctional CD8+ T cells in fibrotic portal tracts, alongside impaired metabolic regulators. PLS-NAFLD was translated into a four-protein secretome signature, which was validated in HCC-naive patients with NAFLD and cirrhosis, while the HCC incidence rates at 15 years were 37.6 and 0% in high- and low-risk patients, respectively. The PLS-NAFLD was modified by bariatric surgery, lipophilic statins and the use of IDO1 inhibitors, suggesting that that it could provide aid in pharmacotherapy and HCC chemoprevention [56] (Figure 1).

### 1.4. Glucagon-like Peptide-1 Receptor Agonists’ Effects on Hepatocellular Carcinoma

There are several studies showing potential favorable effects of GLP-1R agonists on HCC. First, Zhou et al. provided evidence that Ex-4, a GLP-1R agonist, inhibited hepatocarcinogenesis through the cAMP–PKA–EGFR–STAT3 axis [57]. In more detail, Ex-4 attenuated both obesity-dependent and -independent hepatocarcinogenesis, by inhibiting tumor cell growth, potentiating apoptosis. The anti-tumor effects of Ex-4 were associated with high expression of GLP-1R and activation of cAMP and protein kinase A (PKA), while Ex-4 further inhibited the expression of epidermal growth factor receptor (EGFR) and signal transducer and activator of transcription 3 (STAT3), which lie downstream of cAMP-PKA signaling, leading to the suppression of multiple STAT3-targeted genes, amongst which were c-Myc, cyclin D1, survivin, Bcl-2 and Bcl-xl. The tumor-inhibiting effects of Ex-4 were in accordance both in GLP-1R-abundant HCC cell lines and in a xenograft mouse model, wherein both PKA and EGFR had a significant contribution in Ex-4 functions, while Ex-4 treatment led to arrest of cell growth and inhibition of EGFR signaling in both HepG2 and Huh7 cells. In addition, Ex-4 was also demonstrated to downregulate hepatic steatosis in HepG2 human HCC cells, via GLP-1R-mediated activation of PKA [58], while Krause et al. provided evidence that Ex-4 induced autophagy and prevented HepG2 cell regrowth via the modulation of mTOR signaling in HCC [59]. It is worth noting that regarding chronic HCC treatment, Ex-4 did not allow tumor cell regrowth by preventing some resistance mechanisms that the cells can acquire, while it exhibited greater capability to inhibit tumor proliferation than liraglutide, another GLP-1R analogue (Table 2).

Moreover, Yamada et al. demonstrated that GLP-1 significantly suppressed both transforming growth factor (TGF)-α- and hepatocyte growth factor (HGF)-induced migration of HuH7 cells, while it further attenuated the phosphorylation of stress-activated protein kinase/c-Jun N-terminal kinase (SAPK/JNK) by TGF-α and HGF in HCC [60]. Along the same line, Li et al. provided evidence that liraglutide, enhanced the apoptosis of human HCC HepG2 cells in a dose-dependent manner, via increasing the activation of the JNK signaling pathway [61]. Furthermore, another study by Krause et al. demonstrated that liraglutide is a potent candidate for the treatment of HCC, as it had an antiproliferative effect in HepG2 cells inducing autophagy and senescence by the increase of TGF-β1, without altering oxidative stress levels, via the inhibition of the PI3K/Akt/mTOR pathway in HCC [62]. In addition, a study by Kojima et al. showed that a 14-week liraglutide treatment completely inhibited hepatocarcinogenesis in a mouse model of NASH, as it suppressed steatosis, inflammation and hepatocyte ballooning of non-cancerous liver lesions [63]. Finally, it has been demonstrated that natural killer (NK) cells mediate the antitumor effect of liraglutide in HCC, as liraglutide enhances NK cell-mediated oncolytic activity by suppressing the IL-6/STAT3 signaling pathway in HCC [64] (Figure 2).

## 2. Discussion

A wealth of epidemiological data suggests that people with T2D are at an increased risk of developing malignancies [65]. Hyperinsulinemia, which in turn leads to increased levels of insulin-like growth factor, appears to exert negative effects on the microenvironment of tumor cells, as it has been shown to stimulate signaling pathways that inhibit cancer cell apoptosis and promote cancer cell growth, migration and invasion into other tissues [66]. Furthermore, glucose itself increases malignant cell metabolism and proliferation, and elevated serum glucose levels have been consistently associated with increased cancer risk in large observational studies [67]. Therefore, a proportion of the antineoplastic effects of incretin-based therapies could be related to improved blood glucose concentrations and decreased circulating insulin levels. In addition, the prevalence of poor metabolic health has increased worldwide, even amongst normal-weight populations [68]. MAFLD was recently defined as a more accurate description of hepatic disorders with dynamic interactions among environmental and genetic factors and metabolic syndrome components [69], with numerous studies assessing the relationship between MAFLD and mortality or the risk of CV [70].

However, since the early years of commercial use of GLP-1 analogs, there have been concerns about their safety with respect to cancer risk among patients treated with these agents. The uncertainty has been primarily related to animal studies showing that liraglutide treatment in rodents activated GLP-1Rs in C cells and increased the risk of medullary thyroid carcinoma (MTC) [71]. As a result, the use of these drugs in individuals with a personal or family history of MTC or multiple endocrine neoplasia type 2 is contraindicated. Although there is evidence that human neoplastic lesions of thyroid C cells express the GLP-1R [72], robust data on a connection between the risk of MTC and the use of GLP-1 RAs in humans are currently unavailable. Mali et al. have reported that, based on 6,665,794 reports recorded in the European pharmacovigilance database (EudraVigilance), Ex-4, liraglutide and dulaglutide met the criteria to generate a safety signal, which means that thyroid cancer is reported more frequently in relation to these agents compared to other drugs [73]. However, more studies are needed to establish a causal association.

Due to the direct effects of GLP-1 RAs on the pancreas, an increased risk of pancreatic cancer related to long-term therapy with these agents has been postulated. Knapen et al. have shown that the risk of pancreatic malignancy is doubled among current incretin users compared to controls [74]. Those findings were not sufficient to prove the association between incretin therapies and pancreatic cancer, due to the presence of considerable confounding by disease severity and the lack of a duration-of-use relationship. A large meta-analysis that included data from 56,004 individuals enrolled in GLP-1 RA CVOTs showed no association between the use of GLP-1 analogs and the risk of pancreatic cancer or acute pancreatitis [75]. The same conclusion was reached by the meta-analysis by Cao et al. who found that the risk of pancreatic cancer with GLP-1 RA treatment was similar to that observed in the placebo arm of randomized controlled trials [76].

Similarly to the evidence summarized in this review for HCC, preclinical data support that GLP-1 RAs can affect molecular pathways that could protect against other types of cancer. Inhibition of nuclear factor kappa-light-chain-enhancer of activated B cells (NF-κB) activation by Ex-4 has been shown to attenuate breast cancer cell proliferation [77], while inhibitory effects of the same molecule on activation of the ERK-MAPK pathway resulted in limited growth of prostate cancer cell lines [78]. Liraglutide has been shown to regulate the NF-κB signaling pathway and down-regulate ATP-binding cassette subfamily G member 2, and through these mechanisms to exert pro-apoptotic effects in gemcitabine-resistant pancreatic cancer cells [79]. Furthermore, liraglutide and exenatide have been reported to promote malignant cell apoptosis and autophagy by activating the AMPK signaling pathway, thus halting endometrial cancer progression [80]. The molecular effects of GLP-1 RAs could be relevant not only for cancer cell proliferation, but also for the effectiveness of antineoplastic treatments. Wenjing et al. have shown that Ex-4 can mitigate prostate cancer cell resistance to enzalutamide by targeting the PI3K/Akt/mTOR pathway, while the combined use of the two agents appears to improve the inhibitory actions of enzalutamide on tumor cell growth and proliferation and increase malignant cell chemosensitivity [81]. Similar conclusions were reached in the study by Eftekhari et al., who showed that the combination of liraglutide and docetaxel in prostate cancer cells has synergistic effects in terms of treatment efficacy, decreasing the dose of docetaxel, and therefore alleviating systemic toxicity induced by chemotherapy [82].

From a clinical perspective, there are several aspects of the use of GLP-1 RAs in patients with HCC that should be considered and investigated in future studies. First, in the context of liver disease, the pharmacokinetics of several drugs could be altered. However, Flint et al. have shown that liver dysfunction does not increase exposure to liraglutide [83]. On the contrary, the results suggested a decrease in exposure with an increasing degree of liver impairment, providing reassuring data that GLP-1 RAs can be used safely in this population. In a retrospective cohort study that included patients with T2D and cirrhosis, Simon et al. found a lower rate of decompensation events (i.e., ascites, spontaneous bacterial peritonitis, hepatorenal syndrome, hepatic encephalopathy, or esophageal variceal hemorrhage) among GLP-1 RA users compared to patients receiving other glucose-lowering agents, such as sulfonylureas [84]. Although the mechanisms behind these benefits are obscure, the authors postulated that they might be related to improved liver lipid oxidation, the promotion of liver stellate cell quiescence, reduced cellular proliferation, and improved microvascular function. The alleviation of lipotoxicity, glucotoxicity, inhibition of mitochondrial apoptosis, and reduction in liver fat content that have been related to the benefits of GLP-1 RAs in NAFLD might also be important [85]. Finally, the fact that the prognosis liver signature-NAFLD hepatocarcinogenesis risk signature was modified by bariatric surgery and lipophilic statins suggests that it can also be modified by weight loss induction, together with an intervention to reduce lipotoxicity [56]. The potential of this combined approach was recently demonstrated by the extraordinary weight loss achieved by tirzepatide, a GLP-1/GIP receptor dual agonist [86]. Although available data are still limited, tirzepatide has demonstrated impressive clinical benefits in recent trials, including the ability to return more than half of participants who received this agent to normoglycemia [87]. Therefore, whether this new drug can improve outcomes related to HCC, by alleviating the metabolic burden of patients and potentially through additional mechanisms, deserves further evaluation.

GLP-1 RAs are generally very safe agents, with the main adverse effects related to their use being gastrointestinal (GI) disorders. Among them, nausea, diarrhea and vomiting are the most frequently reported, with an estimated incidence of approximately 15–30%, 10–15% and 5–10%, respectively [88]. The available evidence suggests that these side effects are not related to specific agents, but represent class effects, are transient, generally mild to moderate in severity, and could be more intense during dose up-titration [89]. Furthermore, they appear to lead only a small proportion (5–10%) of patients to discontinuation of treatment in clinical trials [90]. However, this percentage might actually be higher in real world settings [91]. Although GI side effects do not constitute an important barrier to treatment in otherwise healthy individuals with T2D, they could be a major problem in patients with malignancies whose general physical status is poor. Therefore, appropriate nutritional strategies and conservative dose escalation may be needed to mitigate risks. Finally, it is known that the weight loss response to GLP-1 RAs presents significant interindividual variability related to genetic factors, sex and clinical characteristics, such as baseline weight [12,92,93]. Although most patients with T2D are overweight or obese, and therefore weight loss promoted by pharmacotherapy is desirable, this may not be the case in people with malignancies who are often characterized by cachexia and poor nutritional status.

## 3. Conclusions

Over the past decade, our knowledge regarding the cellular and molecular mechanisms involved in NASH- and NAFLD-associated HCC development has grown, spotlighting GLP-1 RAs, which had originally emerged as drugs for the management of diabetes, as potential candidates for the treatment of HCC. Animal data suggest that GLP-1 RAs could regulate molecular pathways that are deeply involved in the genesis and progression of HCC, including inflammatory responses, tumor cell proliferation, and oxidative stress. However, future mechanistic studies must assess several aspects of the benefit-to-risk ratio of the use of GLP-1 RAs in patients with HCC, including co-administration with chemotherapy, the incidence of gastrointestinal side effects in a high-risk population, and weight loss management in individuals with poor nutritional status and high rates of cancer cachexia.

## Figures and Tables

**Figure 1 cancers-14-04651-f001:**
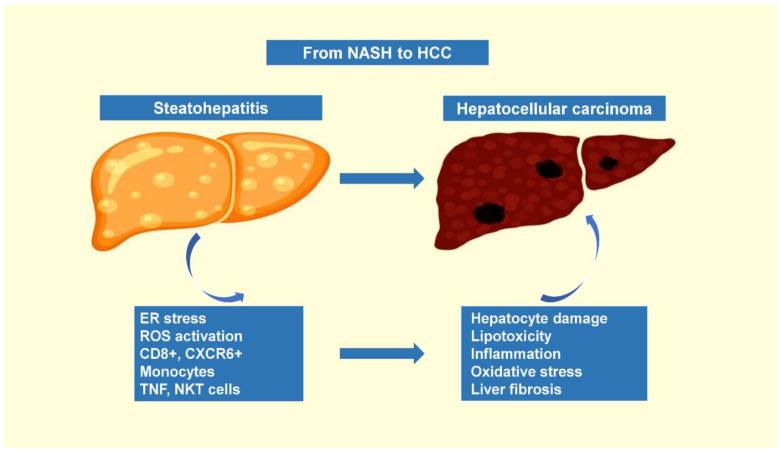
The progression of nonalcoholic steatohepatitis to hepatocellular carcinoma.

**Figure 2 cancers-14-04651-f002:**
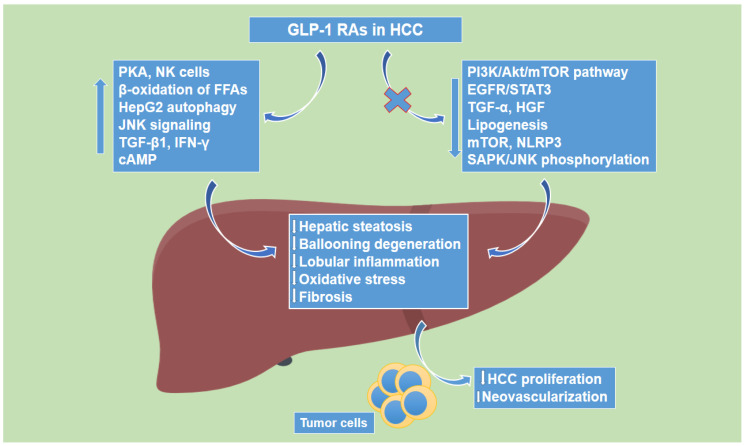
The effect of glucagon-like peptide-1 receptor agonists on hepatocellular carcinoma.

**Table 1 cancers-14-04651-t001:** Summary of studies evaluating the role of GLP-1RAs in steatohepatitis and NAFLD.

Study (Year)	Primary Outcome	Secondary Outcome
Zhu et al. [43] (2021)	GLP-1 RAs reduce intrahepatic adipose tissue, weight mean difference, subcutaneous adipose tissue and visceral adipose tissue	GLP-1 RAs decrease the levels of ALT, AST, body weight, body mass index, waist circumference, fasting blood glucose, HbA1c, TC and TG
Armstrong [44] (2016)	Liraglutide leads to histological resolution of non-alcoholic steatohepatitis without the worsening of fibrosis	Liraglutide treatment is associated with reductions in body weight, BMI and HbA1c
Eguchi et al. [23] (2015)	Liraglutide decreases liver fat deposition, BMI, visceral fat accumulation, AST, ALT, γGT, HbA1c and FPG	Liraglutide reduces the histological features of steatohepatitis, fibrotic stage and NAS score
Ipsen et al. [45] (2018)	Liraglutide reduces NASH progression by diminishing hepatocyte inflammation and ballooning, while it increases hepatic α-tocopherol	Combined liraglutide and chow diet decreases liver weight, TC, LDL-C, VLDL-C and hepatic cholesterol, and increases hepatic vitamin C
Yu et al. [46] (2019)	Liraglutide ameliorates NASH by inhibiting NOD-, LRR- and NLRP3-inflammasome and pyroptosis activation via mitophagy	Mitophagy inhibition with 3-methyladenine/PINK1-directed siRNA weakens the liraglutide-mediated suppression of inflammatory injury
Mantovani et al. [47] (2021)	GLP-1 RAs reduce the absolute percentage of liver fat content and serum liver enzyme levels and leads to histological resolution of NASH without the worsening of liver fibrosis	Treatment with GLP-1 RAs is associated with significant reductions in body weight HbA1c levels
Dong et al. [48] (2017)	GLP-1 RA therapy reduces liver histology scores for steatosis, lobular inflammation, hepatocellular ballooning and fibrosis	GLP-1 RAs treatment significantly reduces the levels of γGT
Rezaei et al. [49] (2021)	GLP-1 RAs therapy reduces ALT, γGT and ALP concentrations	GLP-1 therapy does not alter TG, TC, HDL-C and LDL-C concentrations
Dai et al. [50] (2020)	GLP-1 RAs treatment reduces the liver fat content body weight, waist circumference, ALT and γGT	GLP-1 RAs therapy reduces fasting blood glucose and HbA1c

NASH: nonalcoholic steatohepatitis; GLP-1 RAs: glucagon-like peptide 1 receptor agonists; ALT: alanine aminotransferase; AST: aspartate aminotransferase; HbA1c: hemoglobin A1c; TC: total cholesterol; TG: triglycerides; BMI: body mass index; γGT: gamma-glutamyl transferase; FPG: free plasma glucose; NAS: nonalcoholic fatty liver disease activity score; LDL-C: low-density lipoprotein cholesterol; VLDL-C: very-low-density lipoprotein cholesterol; siRNA: small interfering RNA; NLRP3: NOD-, LRR- and pyrin domain-containing protein 3; ALP: alkaline phosphatase; PINK1: PTEN-induced kinase.

**Table 2 cancers-14-04651-t002:** Summary of studies evaluating the role of GLP-1RAs in HCC.

Study (Year)	Primary Outcome	Secondary Outcome
Zhou et al. [57] (2017)	Ex-4 inhibits obesity-dependent and -independent hepatocarcinogenesis, downregulating EGFR-STAT3 signaling in dose- and time-dependent manners	PKA and EGFR signaling potentiates the tumor-suppressing effect of Ex-4, while GLP-1R expression is enhanced in HCC
Chen-Liaw [58] (2017)	Ex-4 causes partial reduction in triglycerides in steatotic hepatocytes via GLP-1R-mediated activation of protein kinase A	The reduction in hepatocyte triglyceride content is mediated by downregulation of lipogenesis and upregulation of β-oxidation of FFAs
Krause et al. [59] (2019)	Ex-4 induces autophagy and prevents HepG2 cell regrowth via the modulation of mTOR signaling in HCC	Ex-4 decreases HepG2 cells viability and inhibits mTOR expression in a more significant way than liraglutide
Yamada et al. [60] (2021)	GLP-1 attenuates the phosphorylation of SAPK/JNK by TGF-α and HGF in HCC	GLP-1 suppresses both TGF-α- and HGF-induced migration of HuH7 cells
Li et al. [61] (2019)	Liraglutide promotes apoptosis of HCC HepG2 cells by activating the JNK signaling pathway	The proliferation inhibition rate of HepG2 cells increases with time and with the increase in the concentration of liraglutide
Krause et al. [62] (2017)	Liraglutide inhibits cell proliferation in HepG2 HCC cells and induces their autophagy via the inhibition of the PI3K/Akt/mTOR pathway	Liraglutide induces cell cycle arrest and senescence, significantly increasing TGF-β production of HepG2 cells
Kojima et al. [63] (2020)	Liraglutide ameliorates NASH and suppresses HCC formation in diabetic mice	Liraglutide ameliorates steatosis, inflammation, and hepatocyte ballooning of non-tumorous lesions in the liver
Lu et al. [64] (2021)	Liraglutide enhances NK cell-mediated oncolytic activity by suppressing the IL-6/STAT3 signaling pathway in HCC	Liraglutide increases IFN-γ-producing cells in HCC, activating antitumor immunity both in vivo and in vitro

HCC: hepatocellular carcinoma; NASH: nonalcoholic steatohepatitis; mTOR: mammalian target of rapamycin; Ex-4: exenatide; PKA: protein kinase A; GLP-1R: glucagon-like peptide 1 receptor; EGFR: epidermal growth factor receptor; STAT-3: signal transducer and activator of transcription 3; NK: natural killer; FFA: free fatty acid; TGF: transforming growth factor; JNK: c-Jun N-terminal kinase; SAPK: stress-activated protein kinase; IL: interleukin; IFN: interferon; HGF: hepatocyte growth factor; PI3K: phosphoinositide 3-kinase.

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
