# Peer review of "How Far beyond Diabetes Can the Benefits of Glucagon-like Peptide-1 Receptor Agonists Go? A Review of the Evidence on Their Effects on Hepatocellular Carcinoma"

_cancers, 2022, doi:10.3390/cancers14194651_

Round 1

Reviewer 1 Report

Authors reported a detailed comprehensive literature review on this emerging theme of antidiabetic medication effect on NAFLD NASH and liver cancer. I find the description easy to follow and simple to read with minimal jargon and language errors. I find the Figure 1 as bit compressed and authors could increase its height or reduce the width. I find the conclusion too generic and could be focus a little bit on HCC and not include peripheral benefits of GLP-1RA. 

Thanks

Author Response

We thank the reviewer for his/her comments. 

Point 1. We would like to thank the reviewer for his/her criticism. In the revised manuscript, the width and height of the figure has been adjusted so as it doesn't look compressed in the text. 

Point 2. We would like to thank the reviewer for his/her constructive criticism. In the revised manuscript, changes have been made to the conclusion, so that it focuses more on HCC and not on the peripheral benefits of GLP-1RAs.

Reviewer 2 Report

Hepatocellular carcinoma (HCC) is one of the most appearing tumor type and new ideas about its treatment are needed. In this review article the authors discouse potential role of glucagon-like peptide-1 receptor agonists in HCC treatment. In my opinion, this manuscript is "up to date" and well prepared. Supportinf table and figure are present as well. 

1. However, the article concerns also glucagon-like peptide-1 receptor agonists’ effect on nonalcoholic fatty liver disease and steatohepatitis. To make a balance in the structure of manuscript I suggest include additional table (like current table 1) in this section.

2. Moreover, authors can prepare scheme showing summary of information about the progression of nonalcoholic steatohepatitis to hepatocellular carcinoma.

Author Response

Point 1. However, the article concerns also glucagon-like peptide-1 receptor agonists’ effect on nonalcoholic fatty liver disease and steatohepatitis. To make a balance in the structure of manuscript I suggest include additional table (like current table 1) in this section.

Response 1. We would like to thank the reviewer for his/her comments. In the revised manuscript we have added an additional table (Table 1), regarding glucagon-like peptide-1 receptor agonists’ effect on nonalcoholic fatty liver disease and steatohepatitis.

Point 2: Moreover, authors can prepare scheme showing summary of information about the progression of nonalcoholic steatohepatitis to hepatocellular carcinoma.

Response 2: We would like to thank the reviewer for his/her comments. In the revised manuscript, we have added an additional figure (Figure 1), regarding the progression of nonalcoholic steatohepatitis to hepatocellular carcinoma.